# Evaluating the Connection between MicroRNAs and Long Non-Coding RNAs for the Establishment of the Major Depressive Disorder Diagnosis

**DOI:** 10.3390/biomedicines12030516

**Published:** 2024-02-25

**Authors:** Cătălin Prodan-Bărbulescu, Edward Paul Şeclăman, Virgil Enătescu, Ionuţ Flaviu Faur, Laura Andreea Ghenciu, Paul Tuţac, Paul Paşca, Laura Octavia Grigoriţă

**Affiliations:** 1Faculty of Medicine, “Victor Babeş” University of Medicine and Pharmacy Timisoara, 2nd Eftimie Murgu Square, 300041 Timisoara, Romania; catalin.prodan-barbulescu@umft.ro (C.P.-B.); eseclaman@umft.ro (E.P.Ş.); enatescu.virgil@umft.ro (V.E.); .; paul.tutac@umft.ro (P.T.); drpascapaul@gmail.com (P.P.); grigorita.laura@umft.ro (L.O.G.); 2Department I—Discipline of Anatomy and Embryology, Faculty of Medicine, “Victor Babeş” University of Medicine and Pharmacy Timisoara, 2nd Eftimie Murgu Square, 300041 Timisoara, Romania; 3IInd Surgery Clinic, Timisoara Emergency County Hospital, 300723 Timisoara, Romania; 4Department IV—Biochemistry and Pharmacology, Faculty of Medicine, “Victor Babeş” University of Medicine and Pharmacy Timisoara, 2nd Eftimie Murgu Square, 300041 Timisoara, Romania; 5Discipline of Psychiatry, Department of Neurosciences, “Victor Babeş” University of Medicine and Pharmacy Timisoara, 300041 Timisoara, Romania; 6Department III—Discipline of Physiopathology, Faculty of Medicine, “Victor Babeş” University of Medicine and Pharmacy Timisoara, 2nd Eftimie Murgu Square, 300041 Timisoara, Romania

**Keywords:** depression, biomarker, therapy, prognosis, mammalian models

## Abstract

The most prevalent mental illness worldwide and the main contributor to suicide and disability is major depressive disorder. Major depressive disorder is now diagnosed and treated based on the patient’s statement of symptoms, mental status tests, and clinical behavioral observations. The central element of this review is the increased need for an accurate diagnostic method. In this context, the present research aims to investigate the potential role of two non-coding RNA species (microRNA and long non-coding RNA) in peripheral blood samples and brain tissue biopsy from patients with major depressive disorder. This study reviewed the literature on microRNA and long non-coding RNA expression in blood and brain tissue samples in human and animal depression models by retrieving relevant papers using the PubMed database. The results reveal significant variations in microRNA and long non-coding RNA levels in depressed patients, making it a crucial diagnostic tool that predicts treatment outcomes. It can help track severe cases and adjust therapy dosages based on treatment responses. In conclusion, microRNAs and long non-coding RNAs are pertinent biomarkers that can be added to the diagnostic test panel for major depressive disorder. Both microRNAs and non-coding RNAs can also be used as a tool to track patient progress during therapy and to assist the attending physician in tracking the molecular development of the disease.

## 1. Introduction

### 1.1. Major Depressive Disorders

#### 1.1.1. General Considerations

Major depressive disorder (MDD) is one of the most common mental disorders in the general population. While occasional depression is quite common, MDD is characterized by at least two weeks of low mood and greatly affects the quality of life. This illness causes various negative somatic consequences on the human body, which include headaches, exhaustion, increased sensitivity to pain, memory loss, lack of appetite, nausea, diminished libido, high blood pressure, susceptibility to infections, hypersomnia, or insomnia [1]. According to the Institute of Health Metrics and Evaluation, about 280 million people are affected by MDD with a prevalence of 4% in males and 6% in females [2].

#### 1.1.2. Ethiopathogeny

In order to identify serum biomarkers that have diagnostic value in major depressive disorder, it is necessary to mention several theories underlying the occurrence of this disorder. Without knowing the risk factors for this disease, the pathophysiological mechanism cannot be fully understood. Therefore, the description of these risk factors will highlight the pathophysiological chain that further leads to the identification of feasible markers of paraclinical diagnosis.

The monoaminergic idea was the first etiopathogenic concept for MDD. The quantitative malfunction of monoamines underpins this idea (serotonin, norepinephrine, and dopamine). According to the monoaminergic theory, MDD is caused by a lack of these neurotransmitters, particularly in synaptic clefts where nerve impulses are sent. Some brain circuits that connect various brain regions are affected by monoaminergic dysfunction. There are three elements of the triad (Beck’s Triad) that characterize MDD: negative self-interpretation or evaluation; cognitive distortions about the present; and cognitive distortions about the future [3]. MDD is classified with other mental disorders under the label of affective mood disorders. This group brings together several disorders with a common element: a quantitative change in affective mood, which occurs clinically and varies in degree [4]. 

As this review focuses on the role of lncRNA and miRNA in the diagnosis of major depressive disorder, it is significant to note that miRNA changes are important determinants in the risk factors of this condition. For instance: miRNAs play a crucial role in regulating neurotransmitter and neuropeptide systems, and their disruption has been linked to abnormal system performance. It is important to note that all statements made are objective and supported by previous research. In the field of MDD, various miRNAs that affect both monoamine and non-monoamine neurotransmitters have been identified. Serotonin, the primary target of many antidepressants, has been extensively studied in the brain and other tissues for its role in the whole serotoninergic system [5].

A more sophisticated theory had to be developed because the monoaminergic theory failed to explain why antidepressant medication needed a longer time to take effect or why antidepressants were ineffective when simply supplemented with essential amino acids. The latter are precursors of the neurotransmitters thought to be responsible for the onset of depressive symptoms.

As a result, the theory of receptor dysplasticity emerged, shifting the focus from neurotransmitters to their receptors [4]. According to this theory, for instance, in depression, the prolonged deprivation of a particular neurotransmitter’s synapse will result in a compensatory over-activation of the neurotransmitter, which will increase its quantity at the postsynaptic neuron membrane level. Therefore, after neurotransmitters were restored both quantitatively and in terms of the number and function of those receptors returning to normal, the antidepressant effect would manifest [4,6]. Post-mortem studies supporting this notion have demonstrated that there is not a substantial increase in serotonergic receptors in the prefrontal cortex region of suicide victims [7].

Some miRNAs influence neurotransmitter response by affecting receptor density and function, as well as transporter activity. For example, the serotonin transporter (SERT) controls 5-HT neurotransmission by decreasing its concentration in nerve cells’ extracellular fluid [8].

The hypothalamic–pituitary–adrenal axis is the most widely researched neuroendocrine axis that, because of its susceptibility, links childhood psycho-traumas to adult depression. The development of non-responsive hypercortisolemia on the dexamethasone suppression test has demonstrated the malfunctioning of this neuroendocrine axis in depressed patients [1,4]. This hypothalamic malfunction is the source of this neuroendocrine abnormality. As a chronic stress hormone, cortisol has a long-lasting effect and can be neurotoxic, which causes the hippocampus region to atrophy [6,7].

This clarifies the cognitive impairment caused by the focus that is present in depression. Hypothalamic–pituitary–thyroid is a different neuroendocrine axis that has been examined and shown to be abnormal in affective episodes. It does not respond to TRH activation. Endocrine dysfunction linked to affective disease typically manifests as subclinical intensity, accounting for 15% of instances of affective mood disorder [1,4,6]. Aside from thyroid hypofunction, which is typically linked to depressed symptoms, certain studies have also brought attention to instances of subclinical hyperthyroidism. Based on the clinical finding that additional administration of increased doses of thyroxine (T4) may result in remission in those with rapid cycles in whom standard psychopharmacological treatment had no effect, this neuroendocrine axis is particularly significant in bipolar affective disorder [7,9].

In depressed patients, the main biochemical change, in addition to disturbances in monoaminergic neurotransmitters, is represented by an overactive HPA axis. MicroRNAs can influence the activity of the HPA axis by targeting glucocorticoid-related receptors or other pathways [10].

Changes in the expression of the glucocorticoid receptor, its nuclear translocation, cofactor binding, and glucocorticoid-mediated gene transcription may all contribute to the development of glucocorticoid resistance, which causes hyperactivity of the HPA axis [10].

According to this idea, stressful situations are crucial in the occurrence of initial emotional episodes, drawing a parallel with the pattern of epileptic seizures. As the condition worsens, the importance of these stressful life events that set off affective episodes gradually diminishes. Eventually, affective episodes happen on their own without the need for a triggering event because the limbic system becomes more and more sensitized [1,4,6,7].

Substance P was the most researched, and it was believed that its primary function was to inhibit the gene that suppresses brain-derived neurotrophic factor (BDNF) [7,8]. The neurotrophic factor in question plays a crucial role in preserving the viability of neurons. If it were to diminish or become absent in the face of non-specific stress, this would lead to the termination of gray matter atrophy, particularly in the hippocampus, or possibly neuronal death through an apoptotic mechanism [1,4,6]. 

Animal testing and structural neuroimaging studies in depression have shown predilection of certain areas of the brain such as the hippocampus, prefrontal cortex, anterior cingulate gyrus, thalamus, caudate nucleus, nucleus accumbens, and cerebral amygdala. The surface and hypotrophy of the hippocampal regions have both changed. Nonspecific lateral cerebral ventricle enlargement has been observed in certain cases of psychotic depression [4]. Additionally, studies using functional neuroimaging have revealed reduced cerebral metabolism and blood flow, particularly in the right cerebral hemisphere. Clinical manifestations of dopaminergic pathway dysfunction in the nucleus accumbens include fatigue, low energy, and loss of interest and pleasure. The brain regions involved in manic episodes are similar to those involved in depression, but we typically have hyperactivity in these areas [1,4].

#### 1.1.3. Treatment of Major Depressive Disorder

Treatment of major depressive disorder is multimodal. It includes cognitive-behavioral psychotherapy and antidepressant medication [4,6,7,9]. The main classes of psychotropic drugs used in the treatment of depression are selective serotonin reuptake inhibitors (SSRIs), serotonin and norepinephrine reuptake inhibitors (SNRIs), tricyclic antidepressants (TCAs), monoamine oxidase inhibitors (MAOIs), and atypical antidepressants [10,11].

Currently, most clinically effective antidepressants operate on 5-HTergic neurons in the brain, with bupropion being the exception. The brain 5-HT system is the target of antidepressant medication. The antidepressant impact of SSRIs may only be noticeable when inhibitory and excitatory 5-HT receptors are in balance, according to the data on 5-HT receptor density that is currently available [11].

This review evaluates the ability of microRNA and lncRNA to assess the response of patients on antidepressant medication.

Furthermore, no therapy approach has proved substantially successful in stopping relapses or symptom aggravation. Research currently shows that our comprehension of the intricate underlying pathologic situations is still hampered by the deeply ingrained traditional category approach that links only one or a few distinct mediators (often molecular or biological) to an illness-related phenotype. More significantly, the development of individualized and more effective treatments has frequently been impeded by this conventional method [12,13]. As a result, there is a need for a paradigm change and to place an emphasis on the identification of disease trajectories in order to better comprehend the precise evolution of the clinical pattern of various patient populations across time [12,13].

#### 1.1.4. Evaluation of Therapeutic Response

The current therapies for MDD frequently result in unexpected and unpredictable responses. The initial antidepressant that is prescribed will not be effective for more than one-third of the patients [6,7,9]. Out of all treated patients, less than half achieve full remission. According to practice guidelines, the initial clinical response—defined as a 50% decrease in depressive symptoms as assessed by depression rating scales like the Montgomery–Asberg depression rating scale (MADRS) or the Hamilton rating scale for depression (HDRS)—should be evaluated after two to eight weeks of continuous treatment [6,7,9].

Waiting for a prolonged period without outcomes is related to increased distress, disability, and the risk of suicide. It is difficult to develop a valid prognosis for major depressive disorder since clinical indicators are not indicative of treatment outcomes [6,8]. Furthermore, there is currently no known biological marker that can predict therapeutic response for the individual patient before or throughout the early course of antidepressant medication [6,9].

### 1.2. MicroRNA and Long Non-Coding RNA

MicroRNA (miRNA) and long non-coding RNA (lncRNA) are RNA species that are important regulators of normal biological processes, and the abnormal expression of these genes may play a role in the pathogenesis of human diseases such as depression [6].

Of particular importance is the biogenesis and role of these RNA species.

MiRNAs are short, non-coding, single-stranded RNAs ranging in length from 18 to 25 nucleotides that regulate post-transcriptional gene expression. In general, RNA polymerase II transcribes DNA and produces a primary miRNA (pre-miRNA) in the nucleus. The pri-miRNA is then cleaved by ribosomal III Drosha, its cofactor DGCR8, and its cofactor TRBP, resulting in the precursor miRNA. The exportin-5 transporter transports pre-miRNAs to the cytoplasm [6,7]. The cytoplasmic Rnase III Dicer cleaves them into mature miRNAs. Mature miRNAs suppress gene expression by binding to the mRNA’s 3′ untranslated region (3′UTR), resulting in mRNA degradation or translation inhibition [9,11].

A single miRNA may control several target genes, and numerous miRNAs can also regulate the same target gene. This suggests that the production of numerous proteins in cells can work together to enhance their function. Numerous miRNAs are expressed in the nervous system in a tissue-specific manner, such as in axons, dendritic spines, and presynaptic membranes, to participate in neuroplasticity development and progression, including neurogenesis, neuronal maturation, synaptic formation, axon guidance, and neuronal growth [8].

#### 1.2.1. The Role of MicroRNA in Signaling Pathways

MiRNAs are considered important participants in neuronal synaptic communication. Neurotransmitters including glutamate, GABA, dopamine, and serotonin are integrated into presynaptic neurons, packed into vesicles, and supplied to post-synaptic neurons via the synaptic cleft [7]. The encoded receptors present in post-synaptic neurons carry neuronal neurotransmitters, which are encoded by mRNAs that are miRNA targets. MiR-153 controls the synaptosomal-associated protein-25 (SNAP-25), which mediates the calcium-initiated vesicle in conjunction with the plasma layer and neurotransmitter spill in the synaptic cleft [7].

The specialized research indicated that various microRNAs with dysregulated expressions in depressed people targeted genes such as proto-oncogene B-Raf (BRAF), serine/threonine kinase (AKT), and phosphatidylinositol-4,5-bisphosphate 3-kinase catalytic subunit alpha (PIK3CA) [14].

The interaction of target genes and microRNAs revealed previously unknown pathways associated with depression, such as the mammalian target of rapamycin (mTOR) signaling pathway, the neurotrophin signaling pathway, the RAS/RAF/MAPK/ERK signaling pathway, the phosphoinositide 3-kinase (PI3K)/Akt signaling pathway, the mitogen-activated protein kinase (MAPK) signaling pathway, and the signaling pathways regulating stem cell pluripotency [14].

#### 1.2.2. The Influence of MicroRNA on Growth Factors

A class of proteins known as growth factors is responsible for regulating several metabolic processes by triggering and inhibiting downstream signaling pathways [7]. The neurotrophic factor (BDNF), a neurotrophin growth factor involved in synaptic plasticity, synapse formation, and neuronal maturation, was demonstrated by clinical evidence. Additionally, other vascular growth factors like VEGF or VEGFA contribute to the neurotrophic and neuroprotective qualities, respectively [7].

MiRNAs have been shown to directly target the mRNAs responsible for encoding growth factors. It appears that the 3′UTR region of BDNF mRNA, which encodes BDNF, is particularly important for controlling the synthesis of BDNF in neurons in response to stimuli [7].

#### 1.2.3. The Influence of MicroRNAs on Behavioral Phenotypes

Recent research has demonstrated that miRNA modulation affects rodent circadian rhythm. Because circadian rhythm is known to be dysregulated in people with behavioral phenotypes and may be a suitable behavioral phenotype for that disorder, this is especially relevant to the interaction between miRNAs and mood disorders [7]. A microarray platform was utilized to mine miRNAs that were differentially regulated in the adult heads of wildtype and arrhythmic clock mutant cyc01 flies in the Drosophila species. Two potential miRNAs, miR-263a and miR-263b, were constitutively expressed in the cyc01 mutant and exhibited a rhythmic expression pattern in wildtype flies (low during the day and peak at night [7].

#### 1.2.4. Long Non-Coding RNAs

Long non-coding RNAs are the least studied non-coding RNA molecules due to their low production levels and poor sequence conservation. Scientists have been interested in them because of their potential to regulate gene expression. lncRNAs, which are RNA particles longer than 200 nucleotides, do not encode any proteins [4,9].

They are multitasking molecules that regulate DNA methylation, histone alterations, and chromatin remodeling, as well as the expression of genes via transcriptional and post-transcriptional processes [9].

LncRNAs have been shown to interact with mRNAs, acting as miRNA sponges, decoy molecules, or scaffold molecules. As a result, lncRNAs may exist in both the nucleus and the cytoplasm [9].

This review aims to evaluate the existing literature on the correlation between lncRNA and miRNA and their potential as biomarkers for the diagnosis of MDD. It is essential to understand the potential of these non-coding RNAs in providing a biological basis for such a diagnosis, as it could revolutionize the way we approach the diagnosis and treatment of this disorder.

One of the critical questions raised by this review is whether depression can be solely diagnosed through clinical and psychological examinations, or if there are specific biological markers that can help in identifying the diagnosis of MDD. While clinical and psychological assessments are valuable in the matter, the addition of biological markers could provide better methods for diagnostic accuracy. This could lead to earlier detection and intervention, ultimately improving patient outcomes.

Another important consideration is the identification of the appropriate biological fluid or tissue to detect these biomarkers. Blood, cerebrospinal fluid, and brain tissue all help in identifying biomarkers in the case of MDD. Each of these biological samples has its advantages and limitations, and further research is needed to determine which source is the most reliable for detecting lncRNA and miRNA biomarkers of MDD.

The review also explores the potential implications of identifying lncRNA and miRNA biomarkers related to MDD. If specific non-coding RNAs are found to be associated with MDD, it could lead to the development of novel diagnostic tests and therapeutic interventions. Furthermore, understanding the molecular mechanisms underlying MDD could open new avenues for targeted drug development, ultimately leading to more effective treatments for this debilitating disorder.

## 2. Materials and Methods

All the studies included in this review were retrieved from the PubMed database. The following keywords, namely major depressive disorder, microRNA, long non-coding RNA, blood compartments, biomarker, and brain biopsy, were used to search titles, topics, and keywords for relevant literature. In the present review, only original articles written in English were included. In addition, all the references cited in the articles of interest included in this review were also examined to identify further relevant manuscripts. Following the electronic search, identical references were excluded. Based on the inclusion and exclusion criteria, titles, abstracts, and study procedures were carefully screened.

The current study included the scientific papers that quantified miRNAs and/or lncRNAs in patients with MDD as well as animal models of depression.

To determine which studies should be included in the present analysis, the following criteria were used: (1) case–control studies on human subjects with MDD evaluating the expression of miRNA and lncRNA; (2) studies performed on animal models of depression; (3) studies assessing the potential diagnostic of various miRNAs and lncRNAs in MDD; (4) MDD diagnosed using the Structured Clinical Interview for DSM-IV Axis I Disorders (SCID-I); (5) a control group composed of healthy subjects; and (6) articles published in the English language.

The documents that were excluded from the current analysis were as follows: (1) non-original papers, such as conference abstracts, and letters to editors; (2) duplicate studies; (3) papers not written in English; and (4) review articles.

Finally, the present analysis further considered only original research articles that presented data regarding the screening and validation of miRNAs and lncRNAs in MDD from the blood or brain tissue for analysis. The initial search yielded 60 articles. Following a full-text review, 36 studies were excluded based on the above-mentioned criteria, leaving only 24 articles for inclusion in the current review.

For a better understanding of the research strategy of this study, study selection, the inclusion and exclusion criteria, as well as data collection, a flow chart of the study was designed (Figure 1 and Figure 2).

## 3. Results

Regarding the examined miRNA and lncRNAs in MDD or healthy controls, there is significant heterogeneity. Based on the tissue and blood samples taken from both humans and rodents, a vast amount of data has been analyzed and provided in tabular format.

### 3.1. First Set of Articles Reviewed—Evaluation of Blood RNA Level Variations in Human Subjects with MDD and Healthy Human Subsets

Table 1 shows the presence of miRNA and lncRNA changes in studies conducted on batches of patients diagnosed with major depressive disorder. The types of miRNA and lncRNA that were upregulated or downregulated in the blood of these patients are indicated.

Plasma changes in lncRNA levels were observed in both patients diagnosed with major depressive disorder and healthy (controlled) individuals. There was no evidence of a frequently elevated miRNA or a frequently decreased miRNA in the examined group. There was no quantitative preponderance of lncRNA changes, but there were certainly multiple serum changes in the plasma of patients with major depressive disorder.

### 3.2. Second Set of Articles Reviewed—Assessment of RNA Levels in Blood Samples/Brain Biopsy Samples from Human Subjects and Rodent Species

Rodents serve as a common animal model in neuroscience and are crucial in studies that look at how neurons function. Additionally, post-mortem brain tissues, such as the prefrontal cortex and anterior cingulate cortex, have been used as a significant resource to investigate miRNA changes between healthy and depressed subjects to obtain more direct evidence that supports miRNAs as clinical biomarkers of depression. Approximately 50 miRNAs have been identified as abnormally expressed in post-mortem brain tissues from depressed patients to date.

According to Table 2, several miRNA changes mentioned in the literature can be highlighted. These changes were discovered in both peripheral blood biological samples and brain biopsies.

miRNA-124 exhibits quantitative changes common to rodents and humans in brain tissues. The modulation of this miRNA-124 on post-mortem brain tissues is highlighted and reveals the possibility of testing this miRNA-124 using brain biopsy, but this is an unfeasible and invasive method for the diagnosis of depression and is accompanied by multiple risks (hemorrhages, hematomas, infection, brain parenchyma lesions, and cranial nerve damage).

Human volunteers and rodents were used to obtain samples. Multiple increases in miRNA-221 values in biological fluids and tissues from humans and animals are presented in Table 2.

This table shows, in addition to clear quantitative changes, the abundance of this miRNA-221 in several human and animal tissues (prefrontal cortex, hippocampus, cerebrospinal fluid, and serum), which makes the analysis of human cerebrospinal fluid in major depression useful in objectifying miRNA-221 changes.

### 3.3. Third Set of Articles Reviewed—Assessment of RNA Levels in Blood Samples after Antidepressant Medication

In the first study (Table 3), the average number of miRNAs found in blood samples was 385. Escitalopram significantly changed the expression of 30 miRNAs, according to differential expression analysis. The antidepressant therapy severely downregulated 2 miRNAs (miRNA-34c-5p and miRNA-770-5p) and upregulated 28 miRNAs [29].

The quantitative expression analysis revealed that antidepressants changed the expression levels of 30 miRNAs in the blood of MDD patients. Many of these miRNAs may be important for controlling gene expressions in the brain; in particular, 13 of them (let-7d, let-7e, miRNA-26a, miRNA-26b, miRNA-34c-5p, miRNA-103, miRNA-128, miRNA-132, miRNA-183, miRNA-192, miRNA-335, miRNA-494, and miRNA-22n) were previously demonstrated to be important for the fundamental mechanisms underlying brain neuroplasticity and stress response. Table 3 shows that 33 MDD patients were enrolled in the second trial [23], and they received antidepressant medication for four weeks. Twenty patients received a prescription for a serotonin-norepinephrine reuptake inhibitor (SNRI) (duloxetine = 13; venlafaxine = 7), and thirteen patients received treatment with a selective serotonin reuptake inhibitor (SSRI) (escitalopram = 2, fluoxetine = 4, paroxetine = 6, and sertraline = 1) [30].

miRNA-183 and miRNA-212 levels increased considerably following SSRI therapy (*p* = 0.016 and 0.028, respectively); however, only miRNA-16 remains statistically significant after multiple comparison corrections [30].

There was no significant difference in miRNA levels before and after medication administration in patients treated with SNRI (*n* = 20) [21]. Participants treated with duloxetine (*n* = 13) had significantly higher miRNA-183 levels after therapy (*p* = 0.011); however, it was no longer statistically significant after multiple comparison corrections. Venlafaxine-treated patients (*n* = 7) had no significant changes in miRNA levels before and after the treatment [30].

This study indicates that serum miRNA183 and miRNA212 levels dramatically increased following four weeks of antidepressant medication. Significantly higher miRNA-16 levels were identified in the SSRI group but not in the SNRI group, implying that different types of antidepressants may alter different sets of miRNAs [30].

In the third trial, 222 miRNAs were discovered in plasma samples from MDD patients, of which 40 were differently expressed following therapy. Twenty-three miRNAs were found to be considerably overexpressed, with fold increases ranging from 1.85 to 25.42, while 17 miRNAs were found to be significantly downregulated, with fold changes ranging from 0.28 to 0.68. Pathway analysis found 29 upregulated miRNA pathways and 20 downregulated miRNA pathways [31].

All five main pathways (cancer, Wnt signaling, axon guidance, endocytosis, and MAPK signaling) share six dysregulated miRNAs: miRNA-146b-5p, miRNA-146a-5p, miRNA-221-3p, miRNA-24-3p, and miRNA-26a-5p [31].

As a result, the modifications seen following the start of a treatment cycle indicate that the levels of miRNAs vary during therapy. This suggests that the targeted patients can be monitored throughout time using an accurate instrument to measure the variations (Figure 3).

## 4. Discussion

MDD is the most prevalent and challenging mental condition to treat, as well as a hot issue in the fields of psychology and neuroscience [16]. By 2030, major depressive disorder (MDD) is expected to rank as the second most common cause of illness worldwide. Because the etiology of MDD is poorly known, many individuals do not respond to the commonly prescribed treatment [25].

The issues of this study are based on the possibility of correlating plasma or brain miRNAs and lncRNAs in mammalian models with MDD.

A variety of miRNAs and lncRNAs were measured in blood or tissue samples from mice and people as part of the evaluation process. The second consideration was whether these miRNAs and lncRNAs varied in their actions in the following key directions:(1)Do human models of MDD reveal miRNA and lncRNA variations? Are there differences in miRNA and lncRNAs at the plasma level? Do samples of brain tissue biopsy show miRNA and lncRNA variants as well?(2)Do animal models differ in their miRNA and lncRNA composition? Are miRNA and lncRNA variants present in plasma? Do biopsy samples of the brain tissue exhibit miRNA and lncRNA variations?(3)Do human models with antidepressant therapy exhibit miRNA and lncRNA variations?

According to the findings, there are substantial changes in the plasma miRNA and lncRNA levels in individuals with major depression. miRNAs and lncRNAs variability has been found in both human and mouse plasma samples, as well as brain samples. These findings led us to conclude that miRNAs and lncRNAs can be regarded as significant components that can be included in the panel of diagnostic tests for depression while being practicable and quite affordable.

According to one study conducted by Xiao Huang et al., in brain tissues, many miRNAs are highly expressed and may play a role in the degenerative alterations to the central nervous system (CNS) associated with depression. In chronic corticosterone-induced depressed rats compared to normal rats, 26 miRNAs were found to have differential expressions in the prefrontal cortex. These miRNAs regulate genes essential for the stress response and may cause depressive-like behavior through an overactive hypothalamic–pituitary–adrenal axis [9].

Although numerous miRNAs have been found in the brain tissues of rodents used in depressive-like models, only miRNA-124 has so far been found to express itself more frequently in depressed rodents than in normal rodents across various research groups [9]. miRNA-124 is abundant in the brain and may have a role in neuronal differentiation. Prior research has identified miRNA-124 as a potential treatment target and biomarker for serious depression. Nevertheless, the precise mechanism of the action of miRNA-124 in depression is yet unknown.

One specialized study conducted by Wan, Y et al., revealed the fact that miRNA-124 was downregulated in the hippocampus of mice subjected to persistent unpredictable moderate stress. This marker’s heterogeneity may be shown in its abundance in the brains of both humans and rats [16].

Although the exact role of miRNA-124 and its precise mechanism of action is not known, studies reveal that miRNA-124 is a genetic “signature” of major depressive disorder [24].

Another important link in the etiopathogenesis of major depressive disorder is stress. Studies have examined the function of a neuron-specific miRNA, miR-124-3p, the expression of which is significantly dysregulated in stressed mice, using an in vitro system, a rodent model of depression, and a post-mortem human brain [24].

According to Roy et al., it is important to mention that in the rat study, miR-NA-124 showed a response to fluoxetine therapy, giving it a “significant chance” [24].

Another biomarker that enjoys increased heterogeneity is miRNA-221. One study discovered that miRNA-221 expression was unusually high in the cerebrospinal fluid and serum of major depressive disorder patients and the hippocampus of mice, indicating that miRNA-221 might be linked to the onset of depression and can be also used as a clinical biomarker for the molecular diagnosis of depression [16]. Therefore, the panel of paraclinical investigations for depression could include the analysis of cerebrospinal fluid, even if it is not a preferred method, considering the procedural risks, such as infection, hemorrhage, hematoma, paresthesia and even lower limb anesthesia, and post-puncture headache [32].

Furthermore, the researchers discovered that the overexpression of miRNA-221 might increase death in hippocampus neurons. It has been found that hippocampal neuron loss or a reduction in the hippocampus volume is associated with depression. The findings revealed that miRNA-221 was linked to depression [16].

miRNA-221 is also an “agent” that changes in depressed patients undergoing surgery. The combination of physiological and preoperative pathophysiological changes leading to increased stress levels (through the activation of the sympathetic vegetative nervous system and increased plasma levels of norepinephrine and adrenaline) leads to increased plasma levels of miRNA-221-3p [33].

Furthermore, miR-221-3p directly targeted IRF2 to increase astrocyte IFN-α expression. Ultimately, treatment with paroxetine and ketamine affected the expression of miR-221-3p. IFN-α-induced astrocyte activation was reduced by ketamine and paroxetine, likely through the mediation of miR-221-3p [33].

It is important to note that miRNA-124 and miRNA-221 do not respond specifically to therapy for major depressive disorder, limiting their use as biomarkers. However, they can still be used as diagnostic markers for major depressive disorder.

Despite the research on the role of miRNAs and lncRNAs in depression, there is still some controversy. Several examples can be noted. First, poor homogeneity and small sample size in some studies may provide ambiguous and questionable results, which hampers comprehensive assessments of miRNA and lncRNA biomarkers in depression and does not favor clinical use in patients [6].

Secondly, it is noteworthy that there is a significant obstacle in converting basic research findings into clinical applications because several studies have only shown the influence of miRNA and lncRNA on depression in animal or cellular models [6].

Thirdly, the majority of research lacks specificity when evaluating the ncRNA biomarkers’ ability to distinguish depression from other mental conditions, including schizophrenia and bipolar disorder. This information is crucial for the precise diagnosis and management of major depressive disorders [6].

One issue with using these biomarkers currently is that it is unclear whether the expression of miRNAs and lncRNAs in the central nervous system (CNS) correlates with their expression in the peripheral circulation. Additionally, it is uncertain whether peripheral ncRNAs are derived from the CNS in the context of depression. Despite their high expression in the brain and peripheral circulation and their vital role in regulating the pathogenesis of depression, further research is needed to determine the relationship between the CNS and peripheral expression of these biomarkers. This statement highlights the importance of developing suitable biological kits for clinical use [6].

Specialized studies reveal a clear correlation between three miRNAs and major depressive disorder. miRNA-221, miRNA-223, and miRNA-151-3p were directly correlated with major depressive disorder. These miRNAs showed increased values in both blood plasma and cerebrospinal fluid. miRNA-221 showed a stronger increase compared to miRNA-223 and miRNA-151-3p [16,32,33,34].

Another finding is that following antidepressant therapy, various miRNAs and lncRNAs changes have been discovered, indicating that miRNAs and lncRNAs can be used as biomarkers for drug monitoring of antidepressants. It is also possible to follow patients with progressive MDD and to adjust therapy dosages based on molecular response to treatment.

The limitations of this review originate from the studies included in the review, namely relatively limited subject samples, lack of correlation between observations made on animal models and what can be applied to humans, and relatively heterogeneous study groups.

Future directions of research:(1)Possibility of correlating elements identified in rodents with what applies to human subjects.(2)The need to understand and clarify these miRNAs and lncRNAs variations, e.g., what is upregulated and what is downregulated and most importantly which of them are associated with a favorable development of depression.(3)Assess the possibility of using this biomarker in clinical practice.(4)Quantify the human, technological, and financial resources required.

## 5. Conclusions

Numerous miRNAs and lncRNAs reveal abnormal expression in depressed individuals’ brain tissues and/or peripheral fluids, suggesting that they may be involved in the basic pathophysiology of depression.

miRNA-124 demonstrates quantitative alterations in brain tissues that are shared by rodents and humans. The modulation of this miRNA-124 on post-mortem brain tissues is highlighted and reveals the possibility of testing this miRNA-124 via brain biopsy, but this is an unsuitable and invasive method for the diagnosis of depression with numerous risks (bleeding, hematomas, infection, cerebral parenchymal lesions, and cranial nerve injury).

Antidepressants altered the expression levels of 30 miRNAs in MDD patients’ blood, with 13 of them being crucial for controlling gene expressions in the brain. These miRNAs, including miRNA-34c-5p, miRNA-103, miRNA-128, miRNA-132, miRNA-183, miRNA-192, miRNA-335, miRNA-494, and miRNA-22n, are essential for brain neuroplasticity and stress response mechanisms.

miRNA-221 is a biomarker that deserves to be brought into the spotlight because it was found as a biomarker in the cerebrospinal fluid and serum of major depressive disorder patients and rodents. Researchers found that the overexpression of miRNA-221 may increase neuronal death in the hippocampus, which is linked to depression, as hippocampal neuron loss or reduced volume is associated with depression [27].

miRNAs and lncRNAs are practical biomarkers that can be added to the diagnostic panel of testing for major depressive disorder. miRNAs and lncRNAs can also be used as tools to track patient progress during therapy and to assist the attending physician in tracking the molecular development of the disease.

Recognizing these miRNAs and lncRNAs will, therefore, assist in both guiding individualized therapy and understanding the mechanisms of the disease. However, further research is needed to confirm these findings and to identify specific miRNAs that could be used as biomarkers in clinical practice.

## Figures and Tables

**Figure 1 biomedicines-12-00516-f001:**
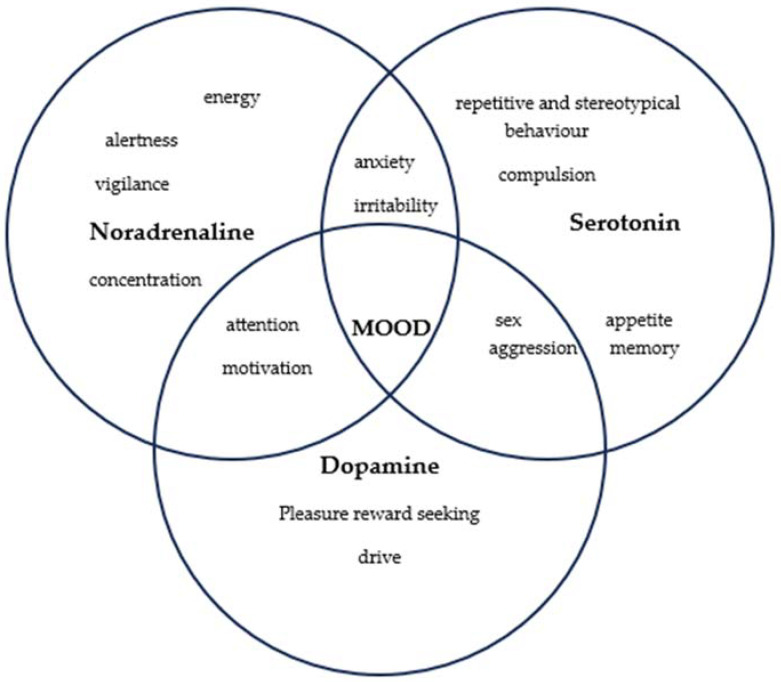
Monoaminergic etiopathogeny for major depressive disorder.

**Figure 2 biomedicines-12-00516-f002:**
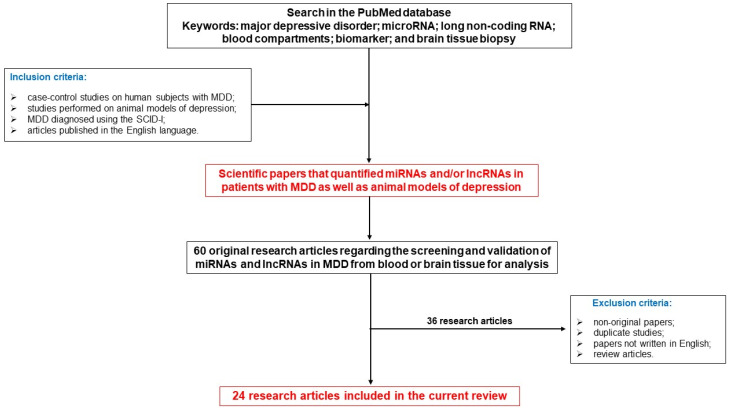
Flowchart of the study.

**Figure 3 biomedicines-12-00516-f003:**
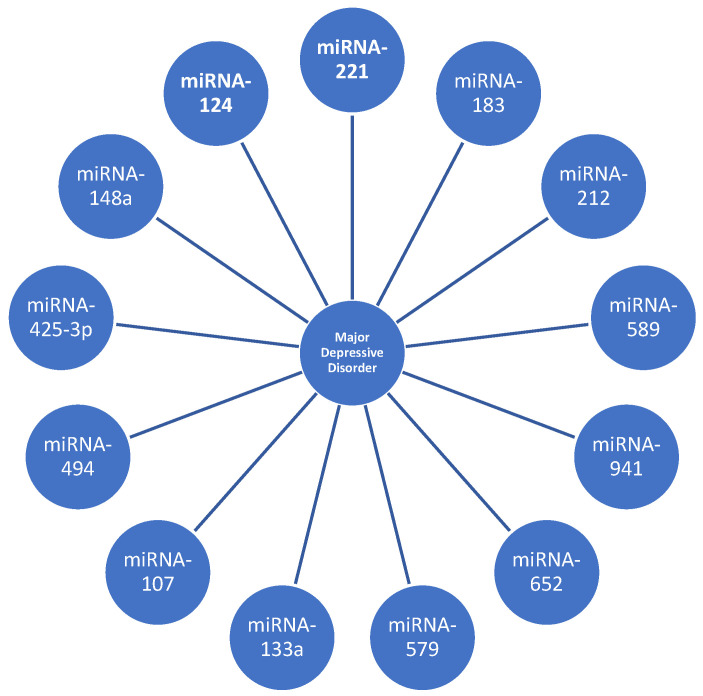
Examples of miRNAs and long non-coding RNAs that are dysregulated in major depressive disorder.

**Table 1 biomedicines-12-00516-t001:** Dysregulation of plasmatic non-coding RNA in major depressive disorder.

Reference	No. of Patients	Controls	Upregulated lncRNA	Downregulated lncRNAs	Total RNA
[15]	16	13	miRNA-107 miRNA-133a miRNA-148a miRNA-425-3p miRNA-494 miRNA-579 miRNA-652 miRNA-941 miRNA-589	miRNA-200c miRNA-381 miRNA-571 miRNA-636 miRNA-1243	9 upregulated and 5 downregulated
[16]	38	27	miRNA-29b-3p miRNA-10a-5p miRNA-375 miRNA-155-5p miRNA-33a-5p miRNA-139-5p	miRNA-106-5p miRNA-590-5p miRNA-185-5p	5 downregulated
[17]	18	18	miRNA-644 miRNA-450b miRNA-328 miRNA-182	miRNA-335 miRNA-583 miRNA-708a miRNA-650 miRNA-654a	4 upregulated, 5 downregulated, and 3 unchanged
[18]	20	20	miRNA-199a-5p miRNA-24-3p miRNA-425-3p miRNA-29c-5p miRNA-330-3p miRNA-345-5p	let-7a-5p let-7d-5p let-7f-5p hmiRNA-1915-3p	6 upregulated, 4 downregulated, and 13 unchanged
[19]	32	32	miRNA-34b-5p miRNA-34c-5p	-	2 upregulated and 3 unchanged
[20]	5	5	534 upregulated RNA	2115 downregulated lncRNAs	534 upregulated RNA, 2115 downregulated lncRNAs
[21]	84	43	miRNA-21-5p miRNA-145 miRNA-223	miRNA-146a miRNA-155 let-7e	3 upregulated and 3 downregulated
[22]	5	2	miRNA-4539 miRNA-4281	miRNA-374b-5p miRNA-98 miRNA-10a-5p	2 upregulated and 3 downregulated

**Table 2 biomedicines-12-00516-t002:** Dysregulations of miRNA-124 in post-mortem brain tissues and blood samples and dysregulations of miRNA-221 in humans and rodents.

**Dysregulations of miRNA-124 in Post-Mortem Brain Tissues and Blood Samples**				
**Reference**	**Species**	**Sample Type**	**miRNA**	**Changes**
[23]	Rodent	Hippocampus	miRNA-124	Decrease
[24]	Humans	The prefrontal cortex (BA46)	miRNA-124	Increase
[25]	Humans	Peripheral blood mononuclear cells	miRNA-124	Increase
**Dysregulations of miRNA-221 in humans and rodent**				
**Reference**	**Species**	**Sample type**	**miRNA**	**Changes**
[26]	Humans	The prefrontal cortex (BA10)	miRNA-221	Increase
[27]	Humans	Cerebrospinal fluid	miRNA-221	Increase
	Rodent	Serum		Increase
		Hippocampus		Increase
[28]	Humans	Serum	miRNA-221	Increase

**Table 3 biomedicines-12-00516-t003:** Correlation between human peripheral blood studies on miRNA expression and the effects of antidepressants.

Reference	Species	Upregulated miRNA	Downregulated miRNA	Therapeutic Agent	Treatment Duration
[29]	Whole-blood sample	miRNA-130b miRNA-505 miRNA-29b-2 miRNA-26b miRNA-22 miRNA-26a miRNA-664 miRNA-494 let-7d let-7g let-7f miRNA-629 miRNA-106b miRNA-103 miRNA-191 miRNA-128 miRNA-502-3p miRNA-374b miRNA-132 miRNA-30d miRNA-500 miRNA-589 miRNA-183 miRNA-574-3p miRNA-140-3p miRNA-335 miRNA-361-5p	miRNA-34c-5p miRNA-770-5p	Escitalopram	12 weeks
[30]	Serum	miRNA-16 (only in selective serotonin reuptake inhibitors) miRNA-183 miRNA-212	-	Selective serotonin reuptake inhibitors or selective serotonin-norepinephrine reuptake inhibitors	4 weeks
[31]	Plasma	miRNA-1193 miRNA-4263 miRNA-3173-3p miRNA-382 miRNA-3154 miRNA-129-5p miRNA-3661 miRNA-1287 miRNA-532-3p miRNA-608 miRNA-3691-5p miRNA-2278 miRNA-3150a-3p miRNA-375 miRNA-3909 miRNA-433 miRNA-937 miRNA-676 miRNA-1298 miRNA-489 miRNA-1909 miRNA-637 miRNA-1471	miRNA-744 miRNA-301b miRNA-27a miRNA-24 miRNA-146a miRNA-126 miRNA-151-5p miRNA-99b miRNA-151-3p let-7d miRNA-221 miRNA-223 miRNA-181b miRNA-146b-5p miRNA-125a-5p miRNA-26a miRNA-652	Escitalopram	12 weeks

## Data Availability

Not applicable.

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
