# Peer review of "Evaluating the Connection between MicroRNAs and Long Non-Coding RNAs for the Establishment of the Major Depressive Disorder Diagnosis"

_biomedicines, 2024, doi:10.3390/biomedicines12030516_

Round 1
Reviewer 1 Report
Comments and Suggestions for Authors
Dear Authors,
I read with interest your work. I think it investigates a promising area of research. However, I have some concerns:
- The Introduction is scattered and fragmentary. Paragraphs from 1.1 to 1.2.5 are not logically related with each other and do not explain to the reader the logical flow of your research. At present stage, Introduction is organized as an explanation of themes related to Depression (etiology, treatment, and so on), but the most important scope of an Introduction should be letting the reader going into a research question. The only sentence related to the manuscript itself is the 238-239, but it does not seem the final consequence of a previous reasoning.
- References of the Introduction are displayed all together at the end of each subparagraph. This does not allow to relate each sentence to the corresponding source. Moreover, your literature search for this section is scarce and too limited: seven references in total. They should be increased with additional search.
- In line 229, what are the numbers 186-187-188 referred to?
- In the Introduction, the depiction of Depression seems too schematic and not considering contemporary research on the topic. I would suggest to take into account the latest works on Network Analysis and Network Intervention Analysis (see, for example, PMID: 36791083, PMID: 38017462), and the reviews that investigated evidence on biological and neuropsychological markers of dysfunction in Depression.
- Methodology: why did you choose to search in PubMed only?
- Figure 1 and Figure 3 have very low quality and should be adjusted with better color and font readability.
- Discussion paragraph is a synthesis of previous results. Instead, it should comment on the findings in light of additional literature. Please change accordingly.
I support the reconsideration of your work after a major revision.
Author Response
Please see the attachment below.

Reviewer 2 Report
Comments and Suggestions for Authors
While the subject of paper by Prodan-Bărbulescu and co-authors is interesting, this article cannot be published in present form since the manuscript needs substantial revision. Some comments needed to be addressed to improve the quality of the manuscript as follows:
Major comments:
1) The introduction needs to be reworked and made more conceptual. All ethiopathogenic factors indicated in the introduction are in fact closely related. A good illustration of this idea is stress-related 5-HT2A receptor dysregulation leading to blunted serotonin turnover (see review Popova et al., 2022; PMID: 35955946). So, authors should indicate in the text and/or graphically the relationship between factors. Next, it is necessary to link the functions of miRNAs (paragraph 1.2) with ethiopathogenic factors of depression. What is known about the regulation of the HPA axis, 5-HT receptors, etc. by miRNAs? In this term paragraph 1.2.3 is almost uninformative.
2) Despite the vast topic covered by the authors in paragraph 1.1., surprisingly few sources are used and they are repeated everywhere (1,4,5,6,7).
3) Little information is provided on the functions of those miRNAs from the blood samples. For at least some, potential targets and regulated signaling pathways are known
4) The authors propose miRNA-124 and miRNA-221 as biomarkers of MDD, but do not discuss the fact that they do not respond to therapy (in the case of miRNA-221, there is only one study). This fact is a limitation for the use of abovementioned as biomarkers and this should be indicated in the text of the manuscript. At the same time miRNA-124 responded to fluoxetine in rats, but this observation from study Roy et al., was not indicated in the discussion section.
Minor comments:
1) The Figure 3 is uninformative at all and should be eliminated.
2) Tables 2 and 3 could be fused in one with subsections. The title of table 2 is incorrect since in the study Roy et al., not only brain samples but also blood serum were studied.
3) I think the rhetorical question (line 488) is inappropriate.
4) In table 4 for reference [24] it is necessary to indicate that the therapeutic agent was escitalopram (at least as indicated in the [24]).
Author Response
Author’s response to Reviewer 2
While the subject of paper by Prodan-Bărbulescu and co-authors is interesting, this article cannot be published in present form since the manuscript needs substantial revision. Some comments needed to be addressed to improve the quality of the manuscript as follows:
We would like to thank you kindly for the added directions and comments. We have made the changes in our article following your comments.
Major comments:
- The introduction needs to be reworked and made more conceptual. All ethiopathogenic factors indicated in the introduction are in fact closely related. A good illustration of this idea is stress-related 5-HT2A receptor dysregulation leading to blunted serotonin turnover (see review Popova et al., 2022; PMID: 35955946). So, authors should indicate in the text and/or graphically the relationship between factors. Next, it is necessary to link the functions of miRNAs (paragraph 1.2) with ethiopathogenic factors of depression. What is known about the regulation of the HPA axis, 5-HT receptors, etc. by miRNAs? In this term paragraph 1.2.3 is almost uninformative.
Thank you very much for your guidance. We consider these indications very valuable for us, so we have made several important changes to the text of our article. We have added correlations between miRNA and etiopathogenic factors of depression. We also considered it very important and relevant to highlight the idea "is stress-related 5-HT2A receptor dysregulation leading to blunted serotonin turnover" (review Popova et al., 2022; PMID: 35955946) together with the citation of this article in the bibliography of the article. We have deleted paragraph 1.2.3, in line with your clarification.
“As the focus of this review is on the role of lncRNA and miRNA in diagnosing major depressive disorder, it is significant to note that miRNA changes are a key factor in the risk factors for this condition. For instance: miRNAs play a crucial role in regulating neurotransmitter and neuropeptide systems, and their disruption has been linked to abnormal system performance. It is important to note that all statements made are objective and supported by previous research. In the field of MDD, various miRNAs have been identified that affect both monoamine and non-monoamine neurotransmitters. Serotonin, the primary target of many antidepressants, has been extensively studied in the brain and other tissues for its role in the whole serotoninergic system [Ortega MA].”
“Some miRNAs influence neurotransmitter response by affecting receptor density and function, as well as transporter activity. For example, the monoamine transporter protein serotonin transporter (SERT) controls 5-HT neurotransmission by decreasing its concentration in nerve cells' extracellular fluid [Gao YN].”
“In depressed patients, the main biochemical change, in addition to disturbances in monoaminergic neurotransmitters, is increased HPA axis activity. MicroRNAs can influence the HPA axis' activity by targeting glucocorticoid-related receptors or other pathways [Ding R].”
“Changes in the expression of the glucocorticoid receptor, its nuclear translocation, cofactor binding, and glucocorticoid-mediated gene transcription may all contribute to the development of glucocorticoid resistance, which causes HPA axis hyperactivity [Ding R].”
“Nowadays, the majority of clinically effective antidepressants operate on 5-HTergic neurons in the brain; bupropion is the exception. The brain 5-HT system is the target of antidepressant medications. The antidepressant impact of SSRIs may only be noticeable when inhibitory and excitatory 5-HT receptors are in bal-ance, according to data on 5-HT receptor density that is currently available [Popova, N.K].”
“This review evaluates the ability of microRNA and lncRNA to assess the response of patients on antidepressant treatment.”
“Furthermore, no single therapy approach has shown to be substantially successful in stopping relapses or symptom aggravation. Research currently shows that our comprehension of the intricate underlying pathologic situations is still hampered by the deeply ingrained traditional category approach that links only one or a few distinct mediators (often molecular or biological) to an illness-related phenotype. More significantly, the development of individualized and more effective treatments has frequently been impeded by this conventional method [Plata-nia GA]. As a result, a paradigm change is required that places an emphasis on the identification of disease trajectories in order to better comprehend the precise evo-lution of the clinical pattern of various patient populations across time [Platania GA].”
- Despite the vast topic covered by the authors in paragraph 1.1., surprisingly few sources are used and they are repeated everywhere (1,4,5,6,7).
Thank you for the indication. We have modified these sources which were repeated in several paragraphs and we have placed the bibliographic sources at the end of the related sentences or phrases.
For instance,
“The hypothalamic-pituitary-adrenal axis is the most researched neuroendocrine axis that, because of its susceptibility, links childhood psycho-traumas to adult depression. The development of non-responsive hypercortisolemia on the dexamethasone suppression test has demonstrated the malfunctioning of this neuroendocrine axis in depressed patients [1,4]. This hypothalamic malfunction is the source of this neuroendocrine abnormality. As a chronic stress hormone, cortisol has a long-lasting effect and can be neurotoxic, which causes the hippocampus region to atrophy [5,6].”
- Little information is provided on the functions of those miRNAs from the blood samples. For at least some, potential targets and regulated signaling pathways are known.
We have enriched the information content on this topic. Thank you very much for the indication.
The specialized research indicated that various microRNAs with dysregulated expression in depressed people targeted genes such as proto-oncogene B-Raf (BRAF), serine/threonine kinase (AKT), and phosphatidylinositol-4,5-bisphosphate 3-kinase catalytic subunit alpha (PIK3CA) [Ferrúa CP].
The interaction of target genes and microRNAs revealed previously unknown pathways associated with depression, such as the mammalian target of rapamycin (mTOR) signaling pathway, the neurotrophin signaling pathway, the RAS/RAF/MAPK/ERK signaling pathway, the phosphoinositide 3-kinase (PI3K)/Akt signaling pathway, the mitogen-activated protein kinase (MAPK) signaling pathway, and the signaling pathways regulating stem cell pluripotency [Ferrúa CP].
- The authors propose miRNA-124 and miRNA-221 as biomarkers of MDD, but do not discuss the fact that they do not respond to therapy (in the case of miRNA-221, there is only one study). This fact is a limitation for the use of above mentioned as biomarkers and this should be indicated in the text of the manuscript. At the same time miRNA-124 responded to fluoxetine in rats, but this observation from study Roy et al., was not indicated in the discussion section.
I have specified those indicated in the text of the manuscript. I hope this meets the requirements. Thank you for your valuable indications.
“It is important to note that miRNA-124 and miRNA-221 do not respond specifically to therapy for major depressive disorder, limiting their use as biomarkers. However, they can still be used as diagnostic markers for major depressive disorder.”
We have also added this element relevant to our theme. Thank you!
According to the study by Roy et al, it is important to mention that in the rat study, miR-NA-124 showed a response to fluoxetine therapy, giving it a "significant chance" [24].
Minor comments:
- The Figure 3 is uninformative at all and should be eliminated.
I understand and have deleted it, thank you!
- Tables 2 and 3 could be fused in one with subsections. The title of table 2 is incorrect since in the study Roy et al., not only brain samples but also blood serum were studied.
We have made the changes according to the indications. We are also merging the two tables into one with two subsections. Thank you for the comment.
- I think the rhetorical question (line 488) is inappropriate.
I understood. I have taken this question out of this paper. Thank you.
4) In table 4 for reference [24] it is necessary to indicate that the therapeutic agent was escitalopram (at least as indicated in the [24]).
I added, thank you for the indications!
We would like to thank you for your indications. We believe that this way we can increase the value of this review, highlight the most relevant aspects, and improve the working method.
Respectfully,
Assistant Profesor Prodan Barbulescu Catalin Flavius
Round 2
Reviewer 1 Report
Comments and Suggestions for Authors
Dear Authors,
my concerns have been addressed. I support the publication of your work.
Author Response
Please see the attachment below.

Reviewer 2 Report
Comments and Suggestions for Authors
The authors responded to most of the comments and significantly improved the quality of the manuscript. I have a few more small comments:
1) Need to improve image quality for Figure 1
2) In Table 3 for [31], replace the text "Antidepressant treatment with Es-citalopram (not re-ported)" with simply "Escitalopram"
3) Although I asked to remove the rhetorical question "Are miRNA and lncRNAs a bridge to personalized medicine in psychiatry?" (line 609), it remained in the text. This is just a question, this is not a research direction. If we carry out research in the proposed directions (lines 602-608), we will obviously be able to understand the importance of micRNAs for personalized medicine.
4) The text of the manuscript requires careful editing. There are various inaccuracies and errors. For example "for diagnosing MDD" (line 305)" instead "for diagnosis of MDD". Some awkward sentences like "the monoamine transporter protein serotonin transporter (SERT)" (line 127). There are other examples, but I just want to draw the attention of the authors to editing.
Author Response
Author’s response to Reviewer 2
The authors responded to most of the comments and significantly improved the quality of the manuscript. I have a few more small comments:
- Need to improve image quality for Figure 1.
Thank you very much for your indications. Due to the fact that the initial editing of figure 1 did not increase its quality enough, we created this figure from the beginning, therefore we hope and trust that the quality now meets your requirements.
- In Table 3 for [31], replace the text "Antidepressant treatment with Es-citalopram (not re-ported)" with simply "Escitalopram".
Thank you for the indication. We have made the correction according to your instructions.

- Although I asked to remove the rhetorical question “Are miRNA and lncRNAs a bridge to personalized medicine in psychiatry?” (line 609), it remained in the text. This is just a question, this is not a research direction. If we carry out research in the proposed directions (lines 602-608), we will obviously be able to understand the importance of micRNAs for personalized medicine.
Please kindly excuse me for the fact that it was not initially removed in the first report you wrote. I have now completely removed this rhetorical question. We understand that this question was not an appropriate one, as multiple aspects are listed above that reveal the role of miRNA in personalized medicine.
Thank you sincerely for the important indication.
4) The text of the manuscript requires careful editing. There are various inaccuracies and errors. For example "for diagnosing MDD" (line 305)" instead "for diagnosis of MDD". Some awkward sentences like "the monoamine transporter protein serotonin transporter (SERT)" (line 127). There are other examples, but I just want to draw the attention of the authors to editing.
I noticed, that there are inconsistencies and irregularities in the text. That is why I have carefully and meticulously corrected the forms of expression. Thank you very much for your guidance.
We would like to thank you for your indications. We believe that this way we can increase the value of this review, highlight the most relevant aspects, and improve the working method.
Respectfully,
Assistant Profesor Prodan Barbulescu Catalin Flavius